# Microstructure and Wear Behavior of Tungsten Hot-Work Steel after Boriding and Boroaluminizing

**Undrakh Mishigdorzhiyn** [1,2,*], **Yan Chen** [3], **Nikolay Ulakhanov** [1] and **Hong Liang** [3]

1   Department of Mechanical Engineering, East Siberia State University of Technology and Management, Ulan-Ude 670013, Russia; nulahanov@mail.ru
2   Institute of Physical Material Science of the Siberian Branch of the Russian Academy of Science, Ulan-Ude 670047, Russia
3   J. Mike Walker '66 Department of Mechanical Engineering, Texas A&M University, College Station, TX 77843-3123, USA; yanchen876@tamu.edu (Y.C.); hliang@tamu.edu (H.L.)
*   Correspondence: druh@mail.ru; Tel.: +7-9148-373-153

**Abstract:** (1) Background: Boron-based diffusion layers possess great application potential in forging and die-casting due to their favorable mechanical and thermophysical properties. This research explores the enhanced wear resistance of tungsten hot-work steel through boriding and boroaluminizing. (2) Methods: Thermal-chemical treatment (TCT) of steel H21 was carried out. Pure boriding was introduced to the substrate through heating a paste of boron carbide and sodium fluoride 1050 °C for two hours. As for boroaluminizing, 16% of aluminum powder was added to the boriding paste. (3) Results: It was shown that boriding resulted in the formation of an FeB/$Fe_2B$ layer with a tooth-like structure. A completely different microstructure was revealed after boroaluminizing—namely, diffusion layer with heterogeneous structure, where hard components FeB and $M_x$ (B,C) were displaced in the matrix of softer phases—$Fe_3Al$ and $\alpha$-Fe. In addition, the layer thickness increased from 105 μm to 560 μm (compared to pure boriding). The maximum microhardness values reached 2900 HV0.1 after pure boriding, while for boroaluminizing it was about 2000 HV0.1. (4) Conclusions: It was revealed that the mass loss during wear test reduced by two times after boroaluminizing and 13 times after boriding compared to the hardened sample after five-min testing.

**Keywords:** thermal–chemical treatment; boriding; boroaluminizing; treatment paste; microstructure; microhardness; wear resistance; hot-work steel

## 1. Introduction

Boriding has been studied for more than half a century. Until recently, the process continues to be developed to satisfy extended fields of applications, such as aerospace, petrochemical refining, textile and food processing, biojoints, or implanting materials manufacturing [1–3]. In comparison to nitriding and carburization, boriding is of limited use. The constraining factors are high brittleness of boride layers and overheating of the substrate metal due to long high-temperature exposure. Researchers have made progress in increasing the plasticity of the layer by means of a single $Fe_2B$ layer formation. The $Fe_2B$ layer can be obtained using low-temperature boriding, ultra-fast boriding in molten electrolyte, direct and alternating current field, concentrated energy flows, etc. [4–17]. Once the brittleness is reduced, boriding acquires potential to be utilized in dynamically loaded machine parts, for instance, to increase surface durability of dies made of hot-work tool steels, which are widely used in forging, die-casting, bending, among others.

Due to the wide application of steel H21 in the aforementioned manufacturing processes, the issue of its surface properties enhancement is of great interest. One approach to improve surface

properties of hot-work tool steel was provided by Chander and Chawla in [18]. The authors proposed duplex/hybrid treatment (plasma nitriding + PVD) as a surface modifying process to enhance wear resistance, hardness, toughness, and fatigue resistance of H21 steel. The plasma-nitrided and duplex CrTiN/AlCrN coatings significantly contributed to the lifetime enhancement of dies during hot forging industrial tests by 55% and 76%, respectively, compared to conventionally heat-treated H21 steel [18]. Despite that, the proposed techniques demand complex equipment, which increases the treatment costs. Besides, one of the major PVD coatings drawbacks is their poor adhesion to the substrate compared to diffusion layers.

The authors in [18] clearly indicate that the single-component thermal-chemical treatment (TCT) methods are surpassed by duplex methods, where at least two elements diffuse into the substrate. The multi-component boron-based diffusion layer has a number of advantages compared to the usual borided layer [19–21]. TCT techniques with boron and a non-metal (boronitriding, borocarburing) or boron and a metal (borochromizing, borotitanizing, boronickelizing, boroaluminizing, etc.) are investigated [22–27]. One of these methods is boroaluminizing (joint saturation with boron and aluminum), which allows increased resistance to wear, high-temperature oxidation, and corrosion [28–30]. Enhancement of the listed properties is provided by the iron borides and aluminides formation, where borides FeB and $Fe_2B$ enhance surface microhardness and wear resistance, while aluminides resist surface oxidation.

Recent studies indicate that the required surface properties can be provided by boroaluminizing in self-protective pastes with heating in muffle furnaces [31]. Elevated treatment temperatures of 1050 °C and 1100 °C adduce to the development of heterogeneous structured layers on top of carbon steels, which are superior to the layers, received at inferior temperatures of 950 °C and 1000 °C in wear resistance. Two-component treatment with boron and aluminum is preferable in comparison to pure boriding due to the potential resistance to oxidation and punching load.

As such, further understanding and approaches in boriding are needed. In the current work, we studied the possibility of improvement of wear resistance of hot-work steel H21 by means of boriding and boroaluminizing. The microstructure evolution and wear behavior of treated samples will be discussed, where the advantages of heterogeneous and tooth-like structure of the layers will be considered with respect to mechanical properties. This research is beneficial for the future design of highly wear-resistant coatings in the prospect of their application in forging and die-casting.

## 2. Materials and Methods

Steel 3Kh2V8F (the analog of AISI H21 steel; for convenience, the latter title will be used throughout the paper later on) was used as a tested material. It is a high-quality hot-work tool steel, alloyed with W, Cr, and V. The full chemical composition of steel is given in Table 1. It is used for manufacturing heavy-duty pressing tools (small inserts of final forming stream, dies and extrusion punches, etc.) during hot deformation of alloyed structural steels and high-temperature alloys, molds for injecting molding copper alloys [32].

**Table 1.** The chemical composition of steel H21, wt %.

| C | Si | Mn | P | S | Cr | W | V | Fe |
|---|---|---|---|---|---|---|---|---|
| 0.26–0.36 | 0.15–0.50 | 0.15–0.40 | up to 0.03 | up to 0.03 | 3.0–3.75 | 8.5–10.0 | 0.3–0.6 | 84.33–87.58 |

A steel billet was machined in rectangular pieces, with nominal dimensions of $20 \times 12 \times 10 \pm 1/2$ mm (length by width by height). Four major faces on each of the samples were ground to a 1000 grit. Finally, samples were rinsed in ethyl alcohol, dried and processed further for TCT under the following conditions.

Diffusion layers were developed using self-protective pastes which comprise sodium fluoride (NaF) as a chemical activator and boron carbide ($B_4C$) as a boriding agent. Al-powder was used in addition to the aforementioned chemical components for boroaluminizing, where the boron carbide

and aluminum ratio was 5:1, respectively. The percentage ratio of treatment paste components is given in the following equations for both methods:

$$B_4C\ 96\% + NaF\ 4\%\ (Boriding) \tag{1}$$

$$B_4C\ 80\% + Al\ 16\% + NaF\ 4\%\ (Boroaluminizing) \tag{2}$$

Some sources indicate the necessity of providing a protective atmosphere while carrying out boriding in treatment pastes [1,33]. On the contrary, conducted experiments have shown the ability of $B_4C$–NaF paste to resist against oxygen penetration. Boron carbide reacts with oxygen forming a fusible boron oxide film on the paste's surface, which protects inner compounds and the steel sample from oxidation:

$$B_4C + 4O_2 \rightarrow 2B_2O_3 + CO_2\ (\Delta G = -2230\ kJ/mol) \tag{3}$$

$$2B_4C + 7O_2 \rightarrow 4B_2O_3 + 2CO\ (\Delta G = -4387\ kJ/mol) \tag{4}$$

The components were pre-kneaded with organic binder and acetone until achieving a slurry composition. The acquired slurry (paste) with the steel samples inside it were placed in cylindrical molds and tamped, as shown in Figure 1. After that, the molds were emptied and the briquettes were drained for one hour at 100 °C in a drying oven. Then, the briquettes with steel samples positioned inside were exposed in the following time-temperature parameters: 2 h exposure and 1050 °C heating in a muffle furnace without a protective atmosphere. It is important to place the briquettes into the preheated at 1050 °C furnace at the beginning of the process. Cooling was conducted in still air at ambient temperature, where the cooling rate was measured as 3 °C/s.

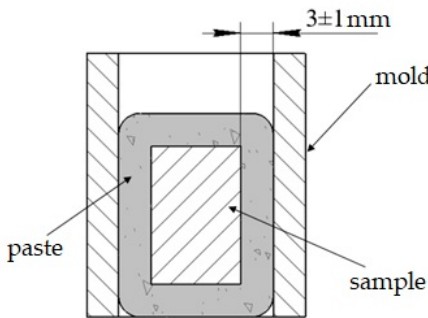

**Figure 1.** Scheme of treatment paste briquettes formation [34].

The metallographic analysis was conducted on scanning electron microscope (SEM) JSM-6510LV (JEOL Ltd, Tokio, Japan) and optical microscope Metam RV-34 (JSC LOMO, St. Petersburg, Russia) using Altami Studio software (Altami LLC, St. Petersburg, Russia). In addition, the atomic force microscope (AFM) Nano-R2 (Pacific Nanotech. Inc., Santa Clara, California, United States) was used to observe the layer zones in order to see the fine-scale morphology. The three-dimensional AFM images were measured in close-contact mode on an area of 25 μm × 25 μm. EDS analysis of the boroaluminized layers was conducted using the INCA Energy 350 (Oxford Instruments plc, Abingdon, UK) microanalysis system installed on the SEM JSM-6510LV (JEOL Ltd, Tokio, Japan) with an acceleration voltage of 20 keV at the Science Center "Progress" of ESSUTM (East Siberia State University of Technology and Management, Ulan-Ude, Russia). It should be noted that carbon content is given to reflect the concentration variations depending on the distance from the surface and does not present actual values. The boron content was evaluated rather qualitatively than quantitatively as well. The final decision on the presence of boron content phases was made based on XRD analysis and microhardness measurement. For the latter, a microhardness tester PMT-3M (JSC LOMO, St. Petersburg, Russia) was used. A pyramid indented with an angle of 136° between opposite faces at a

load of 100 gf was applied with a 10 s for each download, hold, and upload at constant strain rate. The average value of 5 prints was taken to avoid dismissal.

The diffractometer D8 Advance (Bruker AXS Inc., East Cheryl Parkway Madison, WI, USA) was utilized to explore the phase composition in CuKα radiation at the Laboratory of Oxide Systems of BINM SB RAS (Russian Academy of Science, Siberian Branch, Ulan-Ude, Russia). The following parameters were applied to obtain diffraction patterns: signal accumulation of 3 s/point, angular interval 2θ = 5–80°, pitch 0.021°.

The wear test was conducted in accordance with the block-on-ring scheme with dry sliding friction on friction machine SMTs-2 (Figure 2). The sliding speed was 0.8 m/s ($n$ = 300 min$^{-1}$) and the contact pressure (P = 6.86 N/mm$^2$). A hardened disk of high-speed steel R6M5 (analog to AISI M2 high-speed steel) with a hardness of 63 ± 2 HRC was used as a roller (ring). The tested samples were previously borided and boroaluminized in accordance with the aforementioned modes. The bare steel and steel samples after hardening were subjected to the test for comparison. The sample's wear behavior was determined by its mass loss, on an analytical scale with an accuracy of ±10$^{-4}$ g. Due to the significant difference in surface hardness and, consequently, mass loss, it was decided to weight bare and heat-treated steels every minute, and borided or boroaluminized ones every 5 or 10 min. The worn surfaces were examined using SEM after the test finish.

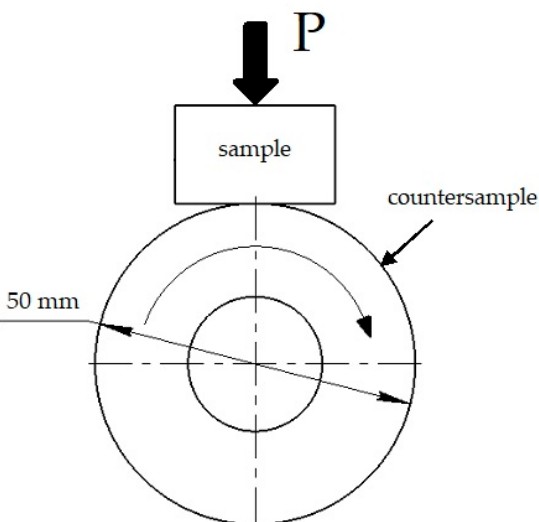

**Figure 2.** The scheme of the wear resistance test.

## 3. Results and Discussion

Boriding in a treatment paste at 1050 °C leads to the formation of layers with a tooth-like microstructure (Figure 3). XRD analysis has shown FeB, BN, and $Cr_2O_3$ presence (Figure 4). The outer iron boride FeB is normally accompanied by $Fe_2B$ as the inner phase on the border with the base metal. The latter phase, probably, was not detected due to insufficient X-ray penetration in the boride layer. $Fe_2B$ presence can be partially confirmed by metallographic and EDS analysis. The presence of BN can be explained by reaction with atmosphere nitrogen:

$$2B + N_2 \rightarrow 2BN \ (\Delta G = -316 \text{ kJ/mol}) \tag{5}$$

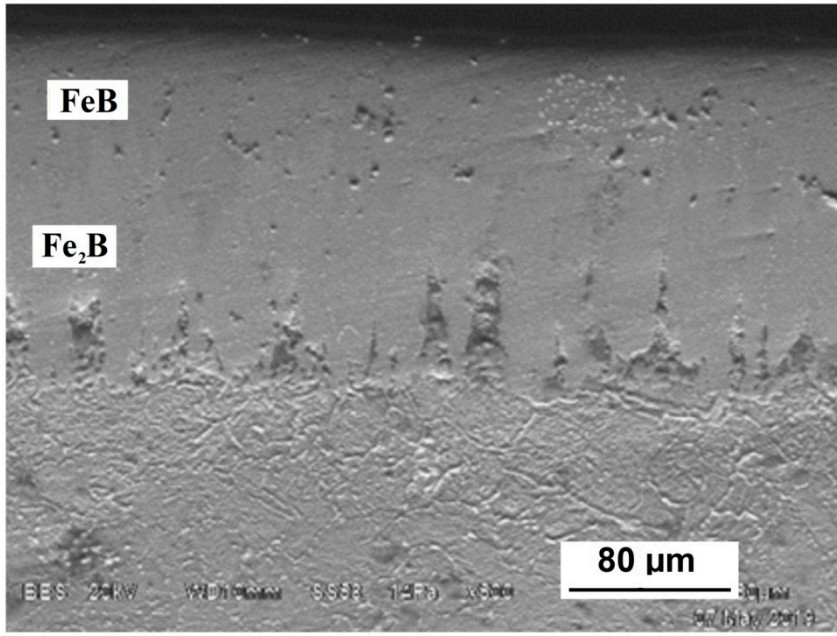

**Figure 3.** SEM image of borided steel H21 after 2-h exposure at 1050 °C.

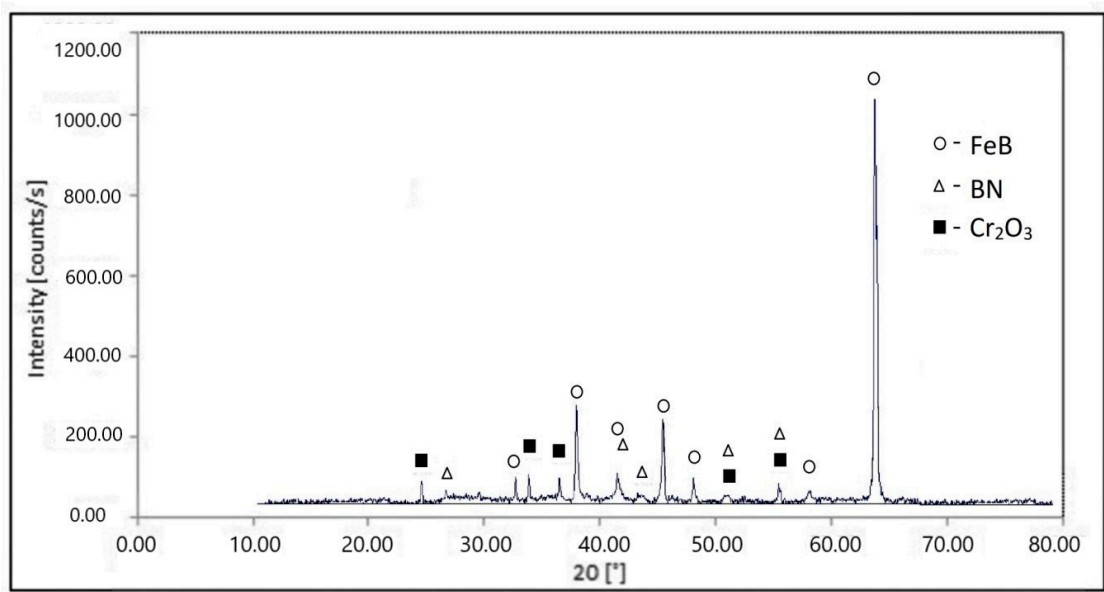

**Figure 4.** XRD pattern of steel H21 after boriding at 1050 °C for 2 h.

Chromium is prone to oxidation above 600 °C. Consequently, $Cr_2O_3$ can form as a result of surface oxidation during high-temperature exposure in the non-controlled atmosphere. Another reason for chromium oxide appearance is its presence in polishing suspension as an abrasive agent. Due to the fact that the samples were firstly investigated on optical microscopy and then subjected to XRD analysis, some of the $Cr_2O_3$ particles (d = 1–2 μm) can penetrate the surface during the cross-section preparation procedure.

EDS analysis has shown that iron borides are highly alloyed with W, Cr, and V (Table 2). For instance, tungsten concentration in the FeB zone is around 8%, while ~10% in the inner-$Fe_2B$ zone. The tungsten-rich zone was revealed below the compound layer. It is formed due to tungsten displacement from the surface by boron, while Cr and V concentration is relatively stable throughout the layer and the base metal. Microhardness measurements have shown unusual high values, which can be referred

to as iron borides alloying with the aforementioned elements. The summarized data are presented in Table 3 in comparison to the literature data.

**Table 2.** The elemental composition of borided steel H21, wt %.

| Distance from the Surface | B [1] | C [1] | V | Cr | Fe | W | Total |
|---|---|---|---|---|---|---|---|
| 25 μm | 12.90 | 6.35 | 0.44 | 2.29 | 69.57 | 8.45 | 100.00 |
| 75 μm | 4.44 | 3.49 | 0.52 | 2.69 | 79.00 | 9.86 | 100.00 |
| 125 μm | - | 9.91 | 0.50 | 2.40 | 73.44 | 13.76 | 100.00 |
| 175 μm | - | 11.61 | 0.55 | 2.50 | 70.80 | 14.54 | 100.00 |
| 225 μm | - | 6.67 | 0.53 | 2.84 | 77.8 | 12.16 | 100.00 |

[1] Boron and carbon content is given to reflect the concentration variation depending on the distance from the surface. The actual values are not possible to define by EDS analysis.

**Table 3.** Some parameters of boride layers after thermal-chemical treatment (TCT) in different treatment pastes.

| Characteristics | $B_4C$-NaF, 1050 °C, 120 min, Steel H21 | $B_4C$-$B_4O_7$-$Na_3AlF_6$ 1000 °C, 20 min, Low-Carbon Steel [33] | $B_4C$-SiC-$KBF_4$ 1000 °C, 20 min, Tool Steels [9] |
|---|---|---|---|
| Layer thickness, μm | 105 | 50 | 50–120 |
| Microhardness, HV0.1 | 2700–2900 | 1800–2100 | 1450–2400 |
| Phase composition | $FeB/Fe_2B$ | $FeB/Fe_2B$ | $FeB/Fe_2B$ |

A diffusion layer with the heterogeneous microstructure characterized by the thickness up to 560 μm was formed after boroaluminizing, which is much thicker than the one after pure boring (Figure 5). The obtained layer can be separated into four structural zones, each on a certain layer depth (Figure 5a). Porous outer Zone 1 contains two types of crystals—the light fine ones and coarse sharp crystals, embedded in a softer matrix (Figure 5b). Zone 2 is placed underneath and characterized by the presence of gray and light crystals with fine or elongated shape displayed in the matrix as in the previous zone (Figure 5c). Zone 3 consists of bainite needles encircled by a light net of the second phase (Figure 5d). The upper zone of the base metal can be designated as a transition sublayer and marked as Zone 4, where the martensitic structure is observed.

Metallographic analysis along with the EDS data has shown that each zone comprises two main phases with peculiar elemental composition and microstructure (Table 4). For instance, aluminum-containing phases such as $Fe_3Al$ (17.5 wt % of Al) and aluminum solid solution in $\alpha$-Fe (10.35 wt % of Al) concentrate in the upper zones and play the role of the soft matrix. The former phase concentrates in Zone 1, while the latter in Zone 2. The XRD analysis confirms the presence of $Fe_3Al$. Besides, other compounds were revealed such as $Fe_2O_3$, FeB, $Fe_7W_6$, and AlN (Figure 6).

The discovered iron oxide $Fe_2O_3$ concentrates in the upper Zone 1 and results in its high porosity. The presence of hematite means that oxygen penetrates through the boron oxide film. At the same time, there were no oxygen traces detected in the inferior zones. Aluminum nitride formed on the surface of the layer as a product of aluminum reaction with atmospheric nitrogen [30,31]. The hard phases in the upper Zone 1 are represented by iron boride FeB, alloyed with Al, W, Cr, and V. Unlike pure boride layer, the crystal growth of iron boride in the (001) direction was changed after two-component TCT. They orient at different angles relative to the diffusion direction (Figure 5b).

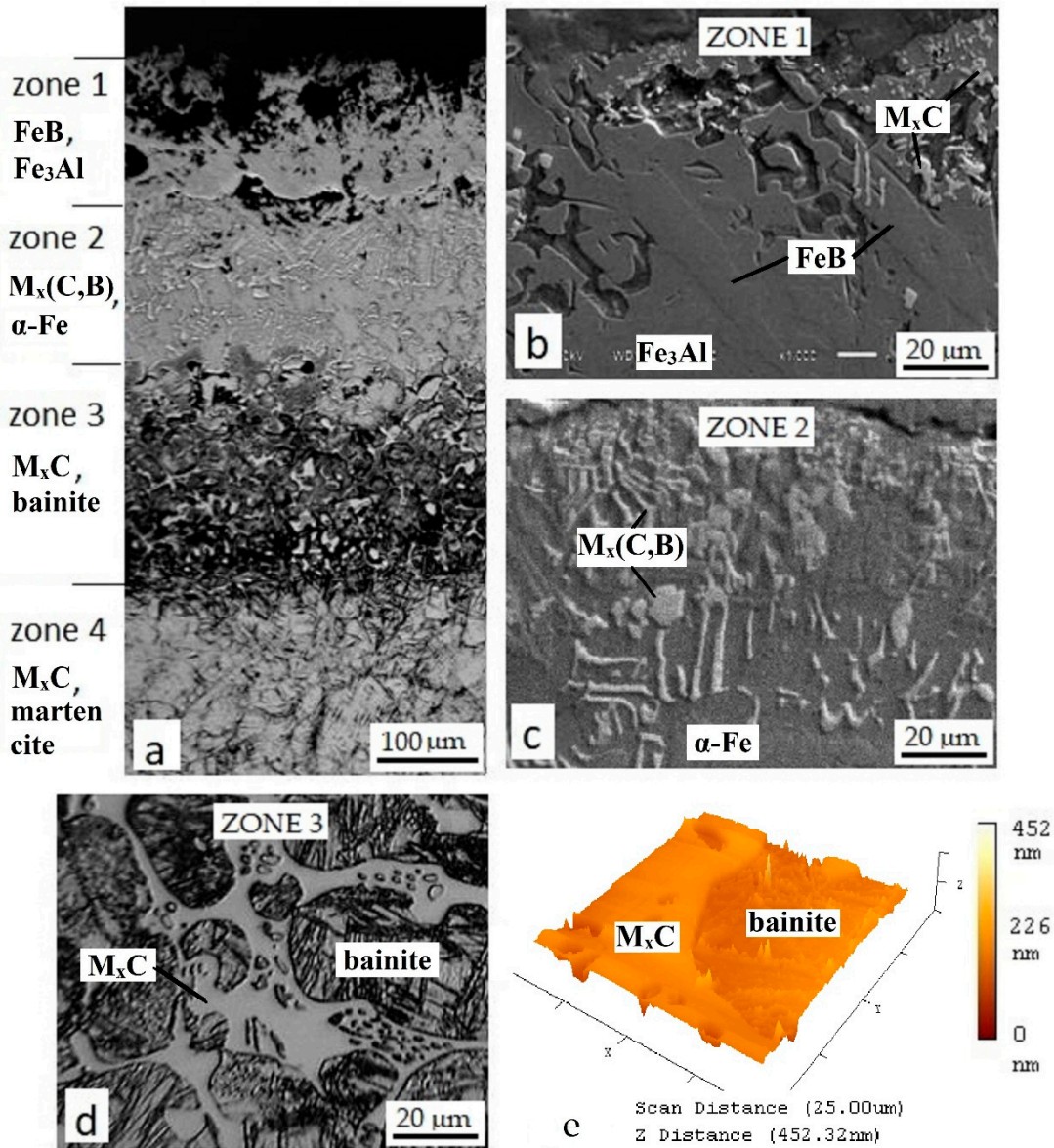

**Figure 5.** Microstructure of boroaluminized steel H21 treated at 1050 °C for 2 h, where (**a**) is the whole layer; (**b–d**) are magnified pictures of the layer zones 1, 2, and 3; (**e**) is the atomic force microscope (AFM) image of Zone 3.

**Table 4.** The elemental composition of boroaluminized steel H21, wt %.

| Distance from the Surface | Probable Phase Composition | B [1] | Al | C [1] | V | Cr | Fe | W | Total |
|---|---|---|---|---|---|---|---|---|---|
| 25–100 µm (Zone 1) | FeB/Fe$_3$Al | 11.50/- | 0.45/17.50 | 2.93/2.22 | 0.64/- | 4.73/0.63 | 73.00/78.52 | 6.75/1.13 | 100.00/100.00 |
| 175–225 µm (Zone 2) | M$_x$(B,C)y/α-Fe(C,Al) | 9.84/- | -/10.35 | 8.60/5.71 | 1.42/- | 1.51/0.34 | 22.39/82.08 | 56.24/1.52 | 100.00/100.00 |
| 375–475 µm (Zone 3) | M$_x$C/bainite | -/- | 1.24/5.11 | 7.22/2.92 | 0.69/- | 4.09/- | 76.65/90.46 | 9.73/1.51 | 100.00/100.00 |
| below 560 µm (Zone 4) | M$_x$C+ martensite | - | - | 7.90 | 0.48 | 2.50 | 80.04 | 9.08 | 100.00 |

[1] Boron and carbon content is given to reflect the concentration variation depending on the distance from the surface and phase composition. The actual values are not possible to define by EDS analysis.

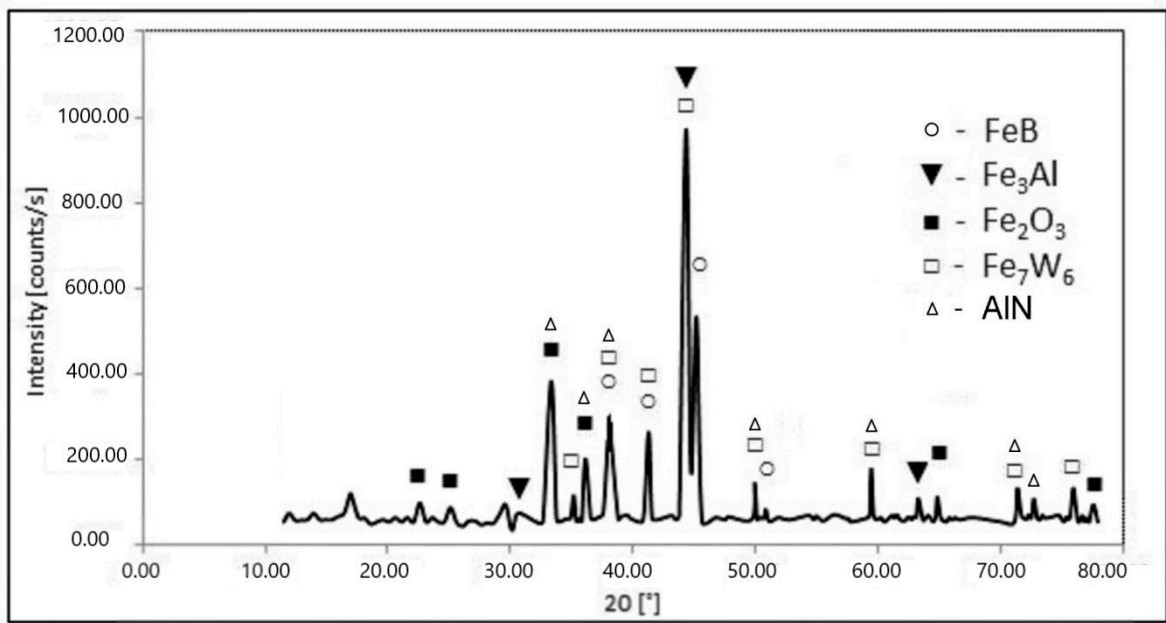

**Figure 6.** XRD pattern of steel H21 after boroaluminizing at 1050 °C for 2 h.

Concerning revealed $Fe_7W_6$, it is known that $Fe_7W_6$ is a high-temperature compound stable at temperature range 1190–1637 °C [35]. Another point is that ferrotungsten main production methods are the carbothermic and carbothermic–silicothermic reduction from tungsten trioxide. Thus, the formation of $Fe_7W_6$ is unlikely. Carbide presence is more realistic instead. The precipitation of $M_xC$ and $M_{23}C_6$ carbides occur during H21 tool steel solidification process and heat treatment, where M is Fe, Cr, and W [27]. It is known that boron can substitute up to 80% of carbon in cementite $Fe_3C$ at 1000 °C [36]. As a result, carboborides $Fe_3$ (C,B) and $Fe_{23}(C,B)_6$ form during the solidification process in the ternary Fe–C–B system. These compounds can be present next to each other and $Fe_2B$ [37]. $M_3(C,B)$, $M_{23}(C,B)_6$ carboborides, and solid solutions ($\alpha$-Fe(Al,C)) represent the phase composition in Zone 2, where M is predominantly Fe and W according to EDS data (Table 4).

Metallographic analysis revealed bainite in Zone 3 (Figure 5d). Its formation can be explained by the presence of undissolved carbides that stimulate the bainite transformation [38]. The second phase in Zone 3 segregates the form of the light net around bainite plates. It is known that H21 steel tends to form a brittle eutectic carbide network [38]. In [39], it was reported that boriding resulted in embrittlement of the H11 steel compared to the non-treated samples. The authors explained such an effect by the fact that the layer itself and the sublayer zone contributed to the bulk embrittlement due to the transcrystalline cleavage. The topography map obtained by AFM analysis shows carbides on the high ground, while bainite is in the lower position (Figure 5e). The element composition of carbides in Zone 3 differs significantly comparing to upper Zone 2. For instance, tungsten content deceased to 9.83 wt %, which is 5.5 times less than in Zone 2. In addition, no boron was revealed by EDS analysis (Table 4).

The microhardness profiles corresponding to the structural zones are given in Figure 7. The profiles differ significantly depending on the TCT method. Thus, boriding has resulted in iron borides formation, with the superior microhardness values of 2700–2900 HV0.1, compared to the values on Armco iron and carbon steels, which lay at an interval of 1800–2400 HV0.1 [9]. The base metal microhardness is almost constant throughout the cross-section and its value is about 550 HV0.1. Microhardness of the base metal is much lower compared to the layer. This high drop in values indicates the absence of the derived transition zone with the variable element and phase composition. That fact could result in the layer chipping under industrial exploitation, especially at shock loads. A completely different profile has been obtained on the boroaluminized samples. Each particular zone possesses certain fluctuation of microhardness values. The high initial peak of about 2000 HV0.1 on

the profile corresponds to the iron boride zone (Zone 1) on top of the layer. Then, the microhardness drops to its minimum value of around 300 HV0.1. Such a great drop corresponds to Zone 2 and relates to its ferritic structure. Despite the presence of hard carbides (some coarse crystals have shown microhardness over 3000 HV0.1) in Zone 2, the average value is 500–700 HV0.1. It is known that tungsten carbides possess high hardness up to 3100 HV0.1 [40]. Another source indicates ternary carbides ($Fe_3W_3C$ and $Fe_6W_6C$) microhardness of 1560 HV0.1 [41]. The low average microhardness in Zone 2 can be explained by the shape and size of carbides. Shallow and/or elongated crystals (up to 20 μm) undergo indentation and displacement in the plastic matrix under a load of 100 gf. Thus, the measurement reflects soft matrix microhardness in this particular zone.

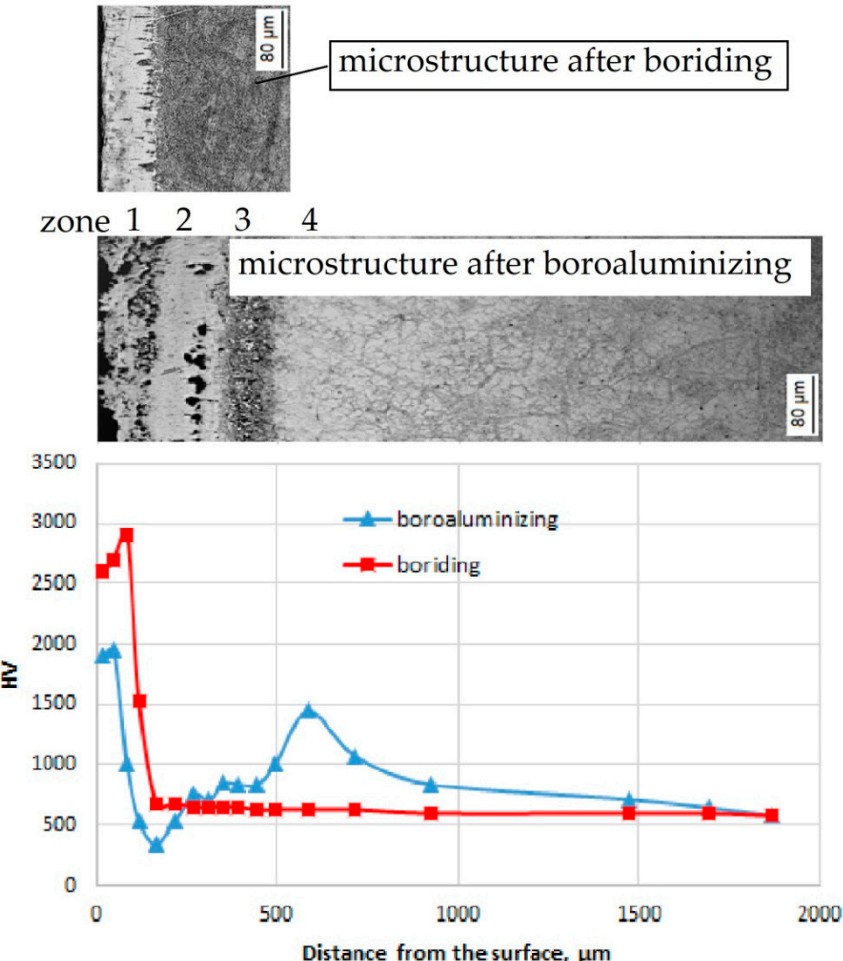

**Figure 7.** Microhardness distribution of boride and boroaluminized layers on steels H21.

The average microhardness in Zone 3 is about 850 HV0.1, where the hardness of bainite is 600 HV0.1, and of the carbide network is about 1400 HV0.1.

The second peak of about 1450 HV0.1 on microhardness profile was measured in transition Zone 4. This growth can be attributed to the carbide-rich zone, which was formed as a result of carbon displacement from the surface. Carbon distribution is a systematic phenomenon for tool steels with a higher alloying level. It is driven by the insoluble nature of carbides in borides [39]. It is known that the enhancement of tungsten concentration in $M_{23}C_6$-carbide can increase its hardness from initial 1080 HV0.1 to 1450 HV0.1 [37].

Beneath the carbide-rich zone, the values gradually decrease from around 1000 HV0.1 to 550 HV0.1. The minimum value coincides with the literature data, indicating 520 HV0.1 after the vacuum heat treatment of H21 steel [18]. The base metal microhardness after boroaluminizing is slightly

superior to the one after pure boriding until achieving a distance of 1.9 mm from the surface. Generally, the microhardness values of the multicomponent layer range significantly and are characterized by sharp rises and drops, which indicate the phase and elemental composition diversity.

It is known that wear and plastic deformation are the main failure processes of hot forging dies [39]. The wear behavior of steel H21 shows that the samples after TCT are superior in wear resistance in comparison with the steels without diffusion layers on the surface (Figure 8). Both heat-treated (quenching followed by tempering) and bare steel have shown critical mass loss since the very beginning of the test and characterized by a sharp wear curve ascent. However, the former is more wear-resistant compared to the non-treated samples and the mass loss was measured as 0.06 g against 0.17 g after the five-min test, respectively. The continuing of the test was decided redundant for these samples due to intense mass loss and high wear rate.

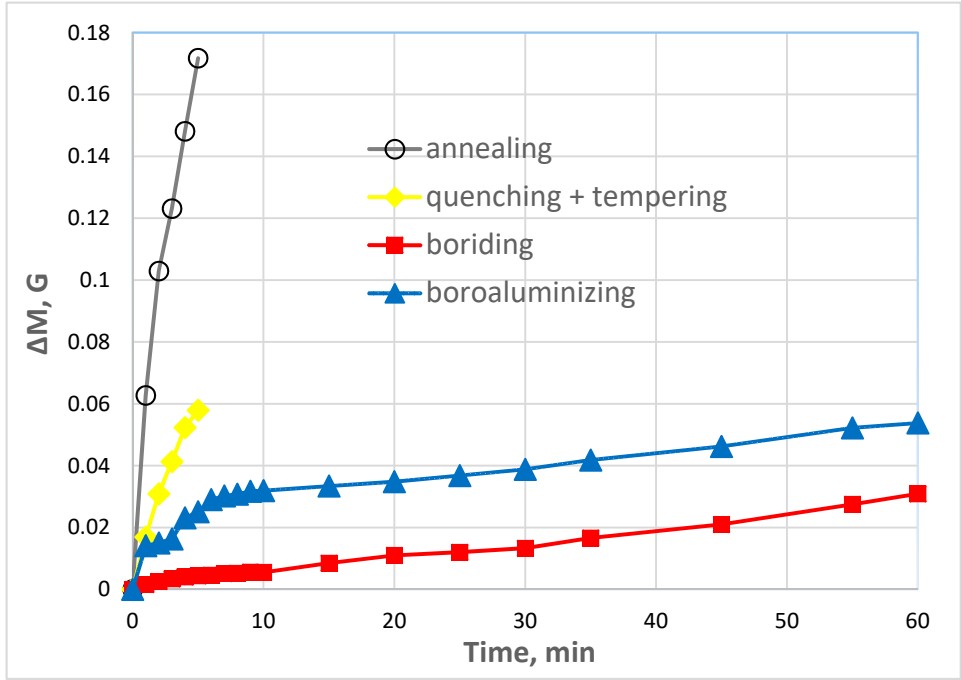

**Figure 8.** Wear resistance of steel H21.

The borided sample has demonstrated the highest resistance to dry friction. In the meantime, the sample after boroaluminizing possesses the corresponding slope of the wear curve after the six-min test. Apparently, the elevated wear rate at the beginning of the test resulted from the presence of porous Zone 1. It is known that high porosity accompanied by the brittle phase presence results in significant wear rate enhancement during abrasive wear without lubrication [42]. As soon as the porous zone is removed, the wear curve slope changed the replicating borided sample's wear profile. It was revealed that the mass loss had reduced by two times after boroaluminizing and by 13 times after boriding compared to the hardened sample after the five-min wear test.

The wear mechanism on steel H21 differs significantly depending on the treatment. The abrasive grooves oriented to the sliding direction are visible on bare and heat-treated steel (Figure 9a,b). Such behavior is inherent in the abrasion wear mechanism [43]. At the same time, local traces of adhesive wear can be observed on bare steel, where abrasive grooves are crumpled or filled with a small number of particles. This indicates the first stage of material transfer, where the shear and rupture of adhesive junctions in asperities of tested material results surface damage.

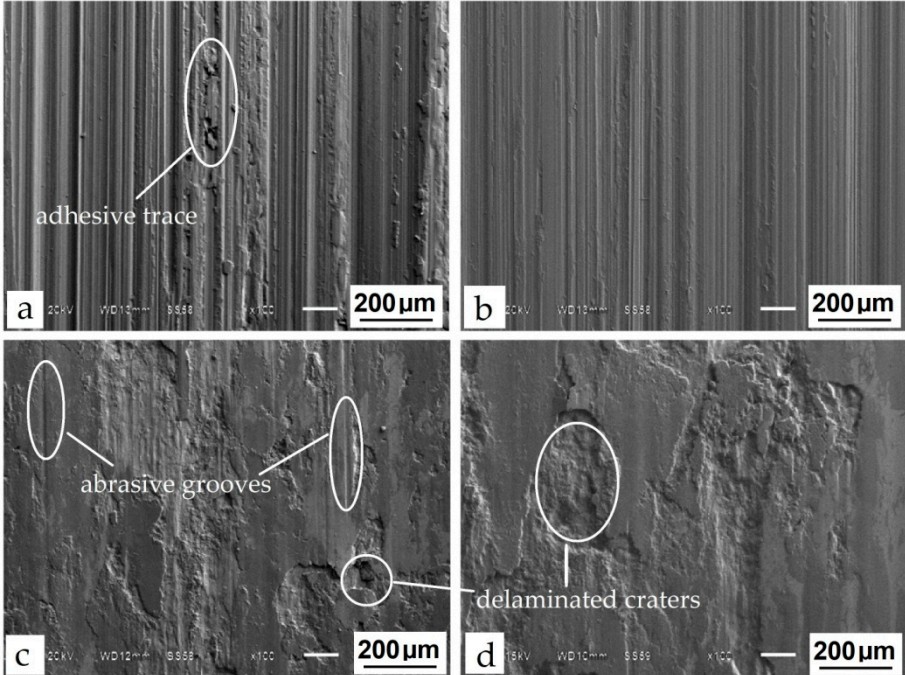

**Figure 9.** SEM images of worn surfaces on steel H21 after wear tests: (**a**) bare steel; (**b**) heat treatment; (**c**) boriding; (**d**) boroaluminizing.

Another type of worn surfaces was obtained after TCT. As can be seen in Figure 9c,d, both borided and boroaluminized samples possess large delaminating and flattened (crumpled) areas. In addition to mentioned damaged surfaces, few separate abrasive grooves parallel to the sliding direction are visible on the surface of the borided sample. In [44], the authors indicated that borided carbon steels are extremely resistant to adhesion and have low welding tendencies due to high iron borides microhardness. This statement can be partially attributed to borided steel H21, wherein a combination of abrasion and adhesion wear takes place. Apparently, the latter process is a predominant wear mechanism in this particular case.

As for the two-component diffusion layer, no abrasive grooves are visible. The damaged surface is characterized by delaminated craters and smeared adhesion lumps. Such a surface was formed as a result of intense adhesive material transfer and its following plastic deformation resulting in the galling build-up. It is known that stainless steel and aluminum possess the most tendencies to gall during contact with the tool surface [45,46]. According to the EDS analysis, the aluminum was content-rich by 17.5% in the upper Zone 1 of the boroaluminized layer, which contributes to its adhesion behavior (Table 4).

EDS analysis of worn surface after wear test has shown a high amount of oxygen on the borided sample (Figure 10a). The lowest oxygen content of about 3.9% was measured in the delaminated crater (Spectrum 1). The rest three spectrums have shown oxygen concentration from 41.4% to 49.8%. This means that oxidation wear takes place in addition to mentioned wear mechanisms. High contact pressure results in friction heating, which accelerates the formation of oxides on wear surfaces. The thickness of the oxide layer is uneven and its lowest value corresponds to elevated tungsten content accompanied by vanadium presence in the vicinity of Spectrum 1. According to [43], the high wear resistance of heat-treated H21 steel is attributed to undissolved carbides $Fe_3W_3C$. Thus, both types of hard compounds-iron borides and carbides $M_xC$ (where M is Fe, W, Cr, V) contribute to wear resistance for borided steel.

The wear surface of the boroaluminized sample was severely oxidized as well (Figure 10b). Oxygen content ranges from 38.9% to 53.4% in points (Spectrums 1–4) and over 55% in an area (Spectrum 5) according to EDS measurements. Aluminum with the content of 2.6%–3.6% was revealed

in addition to alloying metals. Generally, the elemental composition is relatively uniform throughout the wear surface. The oxide layer mainly consists of iron and aluminum oxides, which are considered as tribo-oxides, limiting material transfer during wear [43,44].

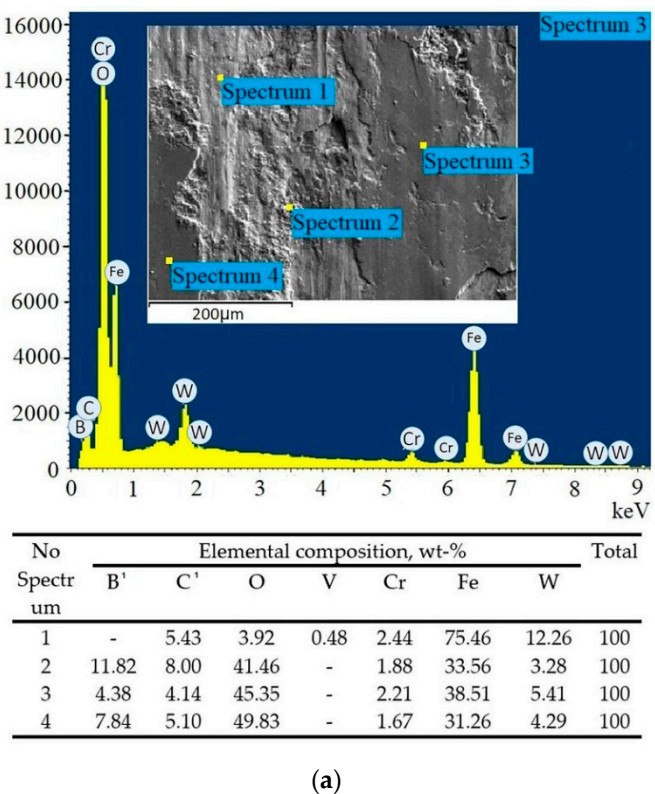

| No | Elemental composition, wt-% | | | | | | | Total |
|---|---|---|---|---|---|---|---|---|
| Spectrum | B[1] | C[1] | O | V | Cr | Fe | W | |
| 1 | - | 5.43 | 3.92 | 0.48 | 2.44 | 75.46 | 12.26 | 100 |
| 2 | 11.82 | 8.00 | 41.46 | - | 1.88 | 33.56 | 3.28 | 100 |
| 3 | 4.38 | 4.14 | 45.35 | - | 2.21 | 38.51 | 5.41 | 100 |
| 4 | 7.84 | 5.10 | 49.83 | - | 1.67 | 31.26 | 4.29 | 100 |

(**a**)

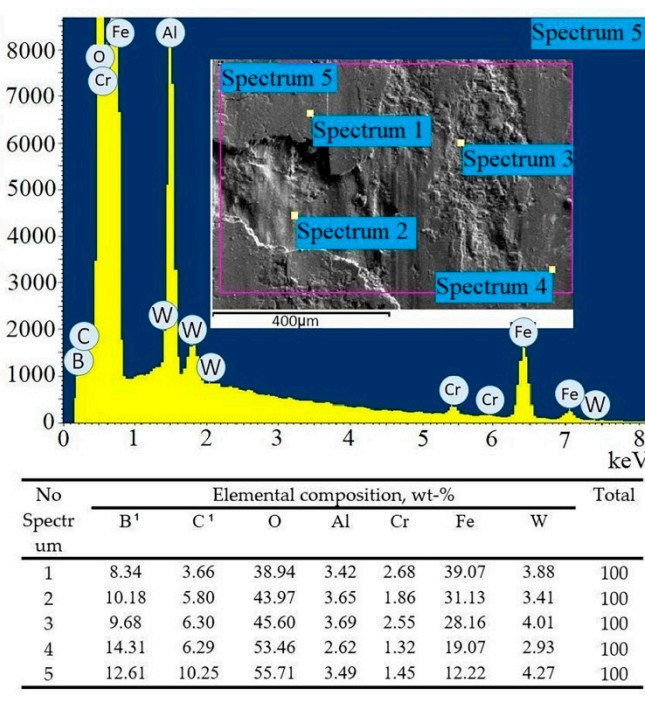

| No | Elemental composition, wt-% | | | | | | | Total |
|---|---|---|---|---|---|---|---|---|
| Spectrum | B[1] | C[1] | O | Al | Cr | Fe | W | |
| 1 | 8.34 | 3.66 | 38.94 | 3.42 | 2.68 | 39.07 | 3.88 | 100 |
| 2 | 10.18 | 5.80 | 43.97 | 3.65 | 1.86 | 31.13 | 3.41 | 100 |
| 3 | 9.68 | 6.30 | 45.60 | 3.69 | 2.55 | 28.16 | 4.01 | 100 |
| 4 | 14.31 | 6.29 | 53.46 | 2.62 | 1.32 | 19.07 | 2.93 | 100 |
| 5 | 12.61 | 10.25 | 55.71 | 3.49 | 1.45 | 12.22 | 4.27 | 100 |

(**b**)

**Figure 10.** EDS analysis results of worn surfaces after wear test on steel H21: (**a**) boriding; (**b**) boroaluminizing.[1] Boron and carbon content is given to reflect the concentration variation depending on the distance from the surface. The actual values are not possible to define by EDS analysis.

## 4. Conclusions

This research generated the following findings: boriding has resulted in FeB/Fe$_2$B layer formation with the tooth-like structure, wherein obtained iron borides are highly alloyed with tungsten and chromium. The two-component thermal-chemical treatment (TCT) results in the surface layer formation with a heterogeneous structure. There are four sublayers (zones) that can be distinguished, each with specific microstructure and properties. The layer thickness has been increased by five times compared to pure boriding and reached 560 μm. The nature of the formation of the heterogeneous microstructure is not established yet. However, it is apparently affected by aluminum presence in the treatment paste.

Microhardness of steel H21 after pure boriding reached up to 2900 HV0.1. Such elevated microhardness is due to iron borides alloying with W, Cr, and V. The microstructure complexity and phase diversity of boroaluminized steel result in microhardness profile variations, where maximum values correspond to iron boride and carbides zones, and minimum to Fe$_3$Al in Zone 2.

Both boriding and boroaluminizing lead to a significant wear resistance improvement of steel H21. However, the wear mechanism of boride and boroaluminized layers differs and largely refers to their microstructure and composition. It has been established that galling is the principal wear mechanism for the two-component layer. On the contrary, the boride layer has a low welding tendency. This property contributes to a higher wear resistance of the borided sample compared to the boroaluminized one. Besides, the tribo-oxide layer forms during the wear test.

**Author Contributions:** Conceptualization, H.L. and U.M.; methodology, N.U. and U.M.; investigation, Y.C., N.U., and U.M.; data curation, U.M.; writing—original draft preparation, U.M.; writing—review and editing, H.L. All authors have read and agreed to the published version of the manuscript.

**Funding:** The study was carried out with a grant from the Russian Science Foundation (Project No. 19-79-10163).

**Acknowledgments:** This work was supported by the Fulbright Visiting Scholar Program. Undrakh Mishigdorzhiyn's internship research results at the Department of Mechanical Engineering, Texas A&M University, in 2018 are used in the paper. The scholar is grateful to Alex Fang, Eugene Chen and Lian Ma of Department of Mechanical Engineering, Texas A&M University, for providing research facilities, consults, and assistance in experiments.

**Conflicts of Interest:** The authors declare no conflicts of interest.

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
