# Peer review of "Microstructure and Wear Behavior of Tungsten Hot-Work Steel after Boriding and Boroaluminizing"

_lubricants, doi:10.3390/lubricants8030026_

Round 1

Reviewer 1 Report

In my opinion, the topic of boride layers has been widely studied by other researchers and the results presented in this manuscript have not contributed additional knowledge. Moreover, there is no sensible practical reason for the work. The layers has been produced  at 1050C; this effectively destroys the prior hardened and tempered microstructure of H21 steel and all of its beneficial mechanical properties (like high strength, excellent hot hardness and fracture toughness).

Author Response

Dear Reviewer, 

Thank you for your comments. 

It is true, boriding has been studied thoroughly. There are a number of prominent schools on Boriding: Campo-Silva (Mexico), Tsipas (Greece), Keddam (Algeria), Kulka (Poland), etc. At the same time, more than 30 papers on the topic are being published annually, which means that boriding is still under high interest. Many of these papers were published by the authors, who previously weren't recognized as experts in boriding. 

In respect to the present study, there are no studies on boriding of steel H21, that fact encourages us to conduct research on this particular steel grade. The practical reason to conduct research was to improve the durability of dies for the bending and hot forging machines.  

Besides, the present study devoted to establishing the regularities of the special microstructure formation on the steel surface, which provides high resistance to mechanical loads. 

Thermal-chemical treatment at 1050C could affect the mechanical properties of steel H21, such as strength, hot hardness, and fracture toughness. This issue should be studied. By now, the evidence of wear resistance improvement has been established clearly. In addition, the post-heat treatment could be applied to heal the core microstructure and provide the necessary mechanical properties of the bulk material. 

Sincerely,

The manuscript authors

Reviewer 2 Report

   The authors investigated thermal and chemical treatment of boronized Tungsten hot-work steels to improve of wear resistance. 

Figure 1. : Indicate the meaning of 1, 2 in Fig. 1 (a), and  1, 2, 3, 4 in Fig. 1(b) in a caption of Figure 1 or in a sentence. 

Figure 8 : Explain the reason of the vibration of HV between 150 -500 Î¼m.

Figure 9. : As for bare steel and quenching + tempering, describe what happened after six seconds later to the sentence.

Author Response

Dear Reviewer,

Thank you very much for your comments.

The response to the comments was provided in the main text. Figure 1. was removed due to another reviewer suggestion to cut down the introduction. The additional explanation was put to Figure 8 and 9. 

Sincerely,

The manuscript authors

Reviewer 3 Report

This paper describes an experimental study of improving the wear resistance of how-work steel using aluminum and boron. The experimental work appears to be well-done and the materials are properly characterized.

The English in the paper needs significant work.

The abstract needs to be abstract. Remove the Background, methods, results and conclusion headings from the abstract and rewrite it as a summary of what was done and what are the key conclusions.

Line 200/201. Is a reference available?

The phase diagram in Figure 7 is very nice.

Please add error bars to Figures 8 and 9 if possible.

Please expand the Discussion section.

Author Response

Dear Reviewer,

Thank you very much for your comments. 

The English in the paper was reworked. The abstract was rewritten according to suggestions. However, the background, methods, results and conclusion headings were left due to editorial advice to use a structured style for the abstract.

The reference was added to the Line 200/201 (line 388/398 in the new version). The Discussion section was expanded.

Sincerely,

The manuscript authors

Reviewer 4 Report

This manuscript by U. Mishigdorzhiyn, Y. Chen, N. Ulakhanov and H. Liang, entitled "Boronized Tungsten Hot-Work Steels for Superior Wear Resistance", highlights the comparative study of microstructure and wear resistance of the tool steel AISI H21 after boronizing and boroaluminizing. Despite that boronizing has been industrially applied for decades, and the corresponding author has recently co-authored multiple papers on the topic of boroaluminizing [1-9], the reviewer hasn't found that the presented results have ever been published before, thus this research may be considered novel. Although boronizing is applied to a relatively small extent to be compared with other thermochemical treatments, such as carburizing and nitriding, the presented results are of a fair interest for engineers and scientists. This fact is indeed evidenced by the large number of up-to-date references from 2016 and later years (13 out of 34). Used experimental methods are generally relevant, the presented results give a comprehensive picture. Illustrations are of a sufficient quality, formatting is generally correct, except for some minor issues. To summarize, this manuscript deserves being published, however, an extensive revision would be required. The reviewer's comments may be found below.

GENERAL CRITISISM.

1. Abstract should be written in a more consise and focused manner (i.e., 'introducing' part about increasing demands should be omitted, the differences between the boronized and boroaluminized layers should be indicated more precisely, the unproven suggestions (about better oxidation resistance, future uses, etc.) should be omitted; abstract could also contain more numbers, giving the idea about the differences between boriding and boroaluminizing, i.e., microhardness increased n times and wear resistance increased m times).
2. Introduction should be rewritten in a clear and brief manner, and be better structured. The Introduction chapter should better indicate the a) necessity for utilizing boronizing, b) advantages and disadvantages of boroaluminizing to be compared with boriding, c) give a brief overview of the state-of-the-art in the field of boriding and boroaluminizing (i.e., what results have so far been obtained). Introduction should also give a better idea of the necessity of the present research. The reviewer also suggests that Introduction could occupy not more than one page.
3. The Discussion chapter is currently too speculative; the suggestions, presented by the authors, should either be based on the experimental results, which are brought in the article, or be confirmed by citing relevant literature data (currently neither is done).
4. The Conclusions chapter is currently too speculative; conclusions, which are not instantly supported by the experimental data and/or its analysis, like about highly alloyed iron borides, recommended applications, etc., should be omitted.
5. English needs revision, especially the terminology (e.g., the term 'composite structure' does not mean that it consists of two or more phases, but that these phases have been separately introduced to this structure or separately synthesized in it, as in, for example, in cemented carbides with ex-situ and in-situ synthesised reinforcement); please, also avoid using terms like 'traditional', 'classic', etc.
6. Measuring units should be written separately from the numerical values.
7. The reviewer recommends to use the en dash to designate all the ranges.

SPECIFIC COMMENTS.

1. Title doesn't reflect the contents of the manuscript and should be revised.
2. Table 1: information about Fe content should also be highlighted.
3. Chapter Materials and Methods: information about microhardness measurements, as well as about atomic force micrsoscopy studies should be provided.
4. Chapter Materials and Methods: please, give the information about sample preparation before thermochemical treatment (descaling, etching, etc.); please, also specify, whether heating during boriding and boroaluminizing was done in air.
5. Lines 156-157: the information about preparation and etching of the samples may be omitted.
5. Line 173: kg/cm2 should be converted to N/mm2 or MPa.
6. Lines 173-174: please, provide here the hardness of the steel R6M5.
7. Line 177: the procedure of measuring the weight loss is unclear; what was the exact dependence between the hardness of the sample and the time periods, after which it was weighted?
8. Figure 4 and Figure 5 have obviously been made, using different equipment (Figure 4 - by a SEM and Figure 5 - by an optical microscope) and thus at different times. It may be seen at Figure 5 that the outer layer exhibited quite an extensive porosity, while no pores may be observed at Figure 4. At the same time, the parameters of boriding and boroaluminizing treatments, given in this article, are identical, the only difference is the absence or presence of aluminium. Therefore the reviewer would like to specify, whether sample, which image is given at Figure 4, was obtained in the frame of the current research, and if it was mechanically treated (polished) after boriding.
9. Lines 187-188: tungsten is prone to oxidation above 650 [deg]C, therefore its lower concentration in the upper part of the borided layer may be indeed explained by its loss due to the reaction with oxygen; was such an opportunity considered?
10. Lines 206-210: it remains unclear, if authors actually meant here that
a) Oxidation of B4C led to efficient protection against oxidation of the substrate steel,
b) Presence of SiC led to efficient protection against oxidation of the substrate steel,
c) Neither factor could save the substrate steel from oxidation.
Please, clarify this.
11. Lines 211-246 and Figure 4: it would be proper to designate the phases on Figure 5.
12. Line 213: actually, three zones may be distinguished in the boronized layer (the 4th zone is the substrate steel).
13. Lines 234-235: what is the evidence that Fe7W6 [citation] "... stabilized by boron during cooling to the ambient temperature ...".
14. Line 241: solid solution can't contant carbides by definition, but carbides and solid solution may compose a mechanical mixture.
15. Lines 243-244: could you, please, specify the principle, how two phases were distinguished, utilizing an atomic force microscope, as well as describe the procedure in this definite case?
16. Figure 6: for the sake of completeness, the analogous XRD chart of the borided steel should also be provided.
17. Figure 7 should be omitted; instead of it, a corresponding reference should be given in the text.
18. Figure 8: for the sake of completeness, the analogous microhardness distribution chart should be provided for the borided sample, as well.
19. According to Figure 8, except for the near-surface microhardness peak, the highest microhardness was obtained beneath the boroaluminized layer, i.e., boroaluminizing worsened the mechanical properties of the substrate steel. Please, comment that.
20. Figure 8 contradicts the information in line 236, as the microhardness in the region, containing the hard (3500 HV) Fe7W6 phase, is only around 300 HV. Please, comment that.
21. Information in lines 265-267 does not refer to the topic of the present research and must be omitted.
22. Lines 271-285: the analysis of the wear mechanisms should be revised; by the reviewer's view, abrasion was the dominant wear mechanism of the untreated and heat treated steel, combination of abrasion and galling wear was the wear mechanism of the borided steel, and galling was the principle wear mechanism of the boroaluminized steel.
23. References:
a) No. 4: please, add the page range.
b) No. 14, 22, 25, 26, 29, 30: please, abbreviate the journal title.
c) No. 19: please, remove 'Vol.'.
d) No. 34: please, add the publication year.

[1] Sizov, I. G., Polyansky, I. P., Mishigdorzhiyn, U. L., & Makharov, D. M. The influence of composition of saturating pastes on the structure and properties of the boron aluminized layer. Obrabotka metallov - Metal Working and Material Science, 2013, 1, 22-25.
[2] Sizov, I., Mishigdorzhiyn, U., Leyens, C., Vetter, B., & Fuhrmann, T. Influence of thermocycle boroaluminising on strength of steel C30. Surface Engineering, 2014, 30(2), 129-133.
[3] Sizov, I., Polyansky, I., & Mishigdorzhiyn, U. The Study of Boroaluminizing in Рastes under Thermocycling and Laser Heating. Advanced Materials Research, 2014, 1040, 907-911).
[4] Polyansky, I., Sizov, I., Mishigdorzhiyn, U., & Butukhanov, V. Improvement of the heat resistance of carbon steels by thermocycling thermochemical treatment with self-protective pastes based on boron carbide and aluminum. In IOP Conference Series: Materials Science and Engineering, 2016, 116(1), p. 012036.
[5] Sizov, I. G., Mishigdorzhiyn, U. L., & Polyansky, I. P. Boroaluminized Carbon Steel. Encyclopedia of Iron, Steel, and Their Alloys. Taylor and Francis, New York, 2016.
[6] Leyens, C., Heinze, S., Schlieter, A., Vetter, B., Polyansky, I., Sizov, I., & Mishigdorzhiyn, U. Thermocyclic Boroaluminizing of Low Carbon Steels in Pastes. Materials Performance and Characterization, 2017, 6(4), 531-545.
[7] Sizov, I., & Mishigdorzhiyn, U. The Influence of Boroaluminizing Temperature on Microstructure and Wear Resistance in Low-Carbon Steels. Materials Performance and Characterization, 2018, 7(3), 252-265.
[8] Mishigdorzhiyn, U., Sizov, I., & Polyansky, I. Formation of Coatings Based on Boron and Aluminum on the Surface of Carbon Steels by Electron Beam Alloying. Obrabotka metallov - Metal Working and Material Science, 2018, 20(2), 87-99.
[9] Mishigdorzhiyn, U. L., & Ulakhanov, N. S. The impact of basic boroaluminizing factors on diffusion layer thickness in low-carbon steels and its mathematical modeling. IOP Conference Series: Materials Science and Engineering, 2018, 411(1), 012049).

Author Response

Dear Reviewer,

Thank very much for the valuable comments. 

Please, find the replies to your comments in the attached file below.

Sincerely,

the manuscript authors 

Round 2

Reviewer 1 Report

The authors have writen " The practical reason to conduct research was to improve the durability of dies for the bending and hot forging machines. "

The paper presents the results of tests on wear resistance against friction. In my opinion, the research should reflect the working conditions of the matrix.

Authors should perform research on fracture toughness of boronized and boroaluminized layers.

There are different graphic design of figures and tables.

The scale in the Fig.3 is unreadable.

Authors should replace 100 g for 100 gf and HV for HV0.1.

The EDS method is not suitable for study of light elements such as carbon or boron. It requires the comments.

There no parameters of SEM for example accelerating voltage that influence on the microanalysis resolution.

Why were pressure and sliping speed fixed at 6,86 N/mm² and 0,8m/s?

The authors didn’t identify all peaks in XRD pattern in Fig.7. These peaks are clearly visible.

Author Response

Dear Reviewer,

Thank you very much for the valuable comments.

Please, find the replies to your comments in the attached file below.

Sincerely,

the manuscript authors

Reviewer 4 Report

This is the revised version of the paper, originally entitled "Boronized Tungsten Hot-Work Steels for Superior Wear Resistance". The reviewer thanks the authors for the careful attention to all the remarks and for the detailed response. By the reviewer's estimation, all the drawbacks of this manuscript have been eliminated, and it is 99.9% ready for publishing. Although the reviewer would still recommend to correct some minor formatting issues, listed below, this may be merged with the proofreading process.

COMMENTS AND SUGGESTIONS.

1. Lines 24-28: for the sake of conciseness, the reviewer recommends to omit the part of the abstract, related to conclusions (as they only summarize the already provided information, but don't add anything new).
2. Line 36: wasn't carburization meant here instead of cementation?
3. Lines 40, 58: 'etc.', not 'etc'.
4. Line 54: 'The ref. [18]' or 'The authors in [18]', not 'The [18]'.
5. Line 56: apparently 'diffusion layer', not 'diffusion layers'; apparently 'usual borided layer', not 'pure boriding'.
6. Line 58: please, insert comma after 'boronickelizing'.
7. Line 59: 'joint saturation with boron and aluminium', not 'joint saturation of with boron and aluminium'.
8. Line 87: apparently 'rinsed', not 'embedded'.
9. Line 90: apparently 'chemical activator', not 'chemical agitator'.
10. Line 105: apparently 'emptied', not 'evacuated'.
11. Line 134: '6.86', not '6,86'.
12. Line 158: please, delete comma after 'shown'.
13. Lines 169, 205, 230: please, split '1050' and '[deg]C'.
14. Table 3, the last column from the left: '1000 [deg]C', not '100 0[deg]C'.
15. Line 181: apparently 'embedded' or 'situated', not 'displaced'.
16. Table 4, second column from the left: 'martensite', not 'martencite'.
17. Line 239: 'chipping under', not 'chipping of under'.
18. Line 240: please, split 'loads.' and 'A'.
19. Line 241: 'fluctuation of microhardness values', not 'microhardness values Vibrations of microhardness'.
20. Lines 242, 243, 245, 246, 251, 253, 259, 347 and Figure 7: the reviewer would recommend to use the designation 'HV0.1', not 'HV' (although it is optional).
21. Line 305: 'steel is extremely resistant' or 'steels are extremely resistant', not 'steel are extremely resistant'.
22. Line 310: apparently 'characterized', not 'exhibited'.
23. Hyphen is used in the references no. 1-3, 8-10, 15, 18-20 in the page ranges, whereas the en dash is used for this purpose elsewhere. The reviewer proposes to use the en dash everywhere for the sake of style uniformity.
24. Reference no. 10: 'pp.' should apparently be deleted.
25. Reference no. 39: an excessive free space should apparently be deleted in front of 'p. 176'.
26. Reference no. 41: please, insert the full stop at the end of the reference.
27. Reference no. 42: please, replace the full stop by comma after '2013, 331'.
28. References no. 41-44: journal titles should be abbreviated. The reviewer suggests the following abbreviations (taken from https://images.webofknowledge.com/images/help/WOS/A_abrvjt.html):
a) Transactions of Nonferrous Metals Society of China -> T Nonferr Metal Soc;
b) Applied Mechanics and Materials -> Appl Mech Mater;
c) Indian Journal of Engineering and Materials Sciences -> Indian J Eng Mater S;
d) Proceedings of the Institution of Mechanical Engineers, Part J: Journal of Engineering Tribology -> P I Mech Eng J-J Eng.

Author Response

Dear Reviewer,

Thank you very much for your valuable comments again.

Please, find the replies to your comments in the attached file below.

Sincerely,

the manuscript authors
